# Assessing Misophonia in Young Adults: The Prevalence and Psychometric Validation of the MisoQuest Questionnaire

**DOI:** 10.3390/healthcare12181888

**Published:** 2024-09-20

**Authors:** Lourival de Almeida Silva, Matias Noll, Gabriel Cunha Siqueira, Alana Karolyne N. Barbosa

**Affiliations:** 1Instituto Federal de Goiás, Goiânia 74270-040, GO, Brazil; 2Instituto Federal Goiano, Campus Ceres, Ceres 76300-000, GO, Brazil; matias.noll@ifgoiano.edu.br

**Keywords:** misophonia, MisoQuest, prevalence, psychometric properties, young adults’ factor analysis, internal consistency, mental health

## Abstract

**Background/Objectives:** Misophonia, characterized by strong emotional reactions to specific sounds, poses significant challenges, particularly in academic settings. This study aimed to examine the psychometric properties of the Brazilian version of the MisoQuest in a sample of high school and university students. The primary objective was to assess its reliability and structural validity to enhance understanding of misophonia in young adults. **Methods:** A total of 549 students (Mean age = 23.2 years, SD = 9.3; 285 females, 260 males, 4 individuals who did not disclose their gender) participated. Both exploratory factor analysis (EFA) and confirmatory factor analysis (CFA) were employed to evaluate the MisoQuest. Fit indices for unifactorial and trifactorial models were compared. Internal consistency was assessed using Cronbach’s alpha. **Results:** The EFA suggested a dominant single-factor structure with high factor loadings (ranging from 0.60 to 0.79). However, the CFA revealed excellent fit for both unifactorial (CFI and TLI = 1.00, RMSEA close to zero) and trifactorial models (CFI and TLI = 1.00, RMSEA = 0.037). The MisoQuest demonstrated high internal consistency (Cronbach’s alpha = 0.93). Additionally, 4.5% of participants were identified as positive for misophonia based on a predefined cutoff score of 61. **Conclusions:** The Brazilian version of the MisoQuest is a reliable and valid tool for assessing misophonia. These findings suggest that the instrument may capture multiple dimensions of the disorder. Given the observed prevalence of misophonia and its impact on students, early identification and tailored interventions are crucial for providing adequate support. Further research is needed to refine the tool and expand its clinical utility.

## 1. Introduction

The term misophonia, derived from the greek μισοφωνία (hatred of sound), was coined in audiology studies to denote a strong aversion to specific sounds, triggering broad negative emotional reactions [1,2,3]. Recently, the Delphi study conducted by an expert panel reached a consensus on the definition of misophonia, emphasizing its characterization by intense negative emotional reactions to specific sounds, which are disproportionate and cause significant distress or impairment [4]. The most bothersome sounds typically originate from the mouth and nose, including chewing, crunching, lip smacking, breathing, snoring, and nasal suction. Pen and mouse clicks and keyboard typing sounds are also considered misophonic triggers [2,3,5]. These sounds evoke neurophysiological and behavioral reactions often misunderstood by peers, friends, and family [6,7]. Common responses to trigger sounds include irritation, anger, disgust, annoyance, sadness, physical reactions, and loss of control [2,6,8]. However, these reactions depend more on psychological profiles, prior experiences, and contextual factors than on sound intensity or frequency [2,3,5]. Initially, misophonia was confused with other abnormalities, such as phonophobia; however, in most cases, typical fear reactions, such as extreme anxiety, are not observed [2,6,8,9]. Although the literature reports various reactions, such as extreme anxiety, compulsive disorders, depression, and aggression, there is no consensus among researchers regarding their occurrence in patients solely diagnosed with misophonia [6].

Concerning the manifestations of the disorder, the first symptoms emerge during childhood and worsen during adolescence [2,8,10]. Emotional instability during this phase often leads to reduced tolerance to trigger sounds, causing difficulties such as social withdrawal, learning challenges, and family conflicts [2,11]. While adults show increased tolerance to trigger sounds, the disorder persists throughout life [2,12,13,14,15]. Moreover, the misophonic spectrum ranges from the subclinical, mild, and moderate forms to the less common severe form [6,8,13]. Individuals with mental disorders may exhibit misophonia symptoms, suggesting a correlation [8,15,16,17]. A study on phenotyping misophonia revealed significant associations with various psychiatric disorders, including anxiety, depression, and obsessive compulsive disorder, as well as medical conditions such as gastrointestinal issues and tinnitus [18]. However, individuals with these disorders may exhibit misophonia symptoms without the reverse being true. This indicates that misophonic individuals can be clinically diagnosed independent of comorbidities [8].

Epidemiological studies on misophonia are scarce and lack sufficient data to estimate its global occurrence. It has been suggested that misophonia prevalence in the general population may be approximately 3.0% [19,20]. Nevertheless, in a study conducted in Ankara, Turkey, Kilic et al. examined a random sample representative of the general population, consisting of 541 individuals aged 15 and older, and reported a prevalence of 12.8% [12]. In a similar study, Vitoratou et al. examined a large, representative sample of the UK general population and estimated the prevalence to be 18% [16]. In Germany, Jakubovski investigated a sample of 2519 individuals from the general population and found a prevalence rate of 5%. [15]. However, specific population studies have reported a high prevalence. For instance, a study on medical students in the United Kingdom found that 49.1% exhibited clinically significant misophonic symptoms [13]. Another study of Turkish high school and university students reported that 13.8% had moderate to severe misophonia, and 41.8% had mild clinical presentations [7]. Zhou et al. investigated 415 university students in Shanghai and observed that 20% of individuals complained about misophonic sounds [17]. Similarly, Brennan et al. identified a prevalence of misophonia ranging from approximately 8% to 20% in a sample of 1084 respondents from a total of 12,131 undergraduate and graduate students aged 18 to 25 [21]. As of May 2024, no studies on misophonia prevalence in South American populations exist.

Therefore, this study aimed to evaluate misophonia prevalence in students from integrated technical and higher education levels at a Brazilian federal institute using the MisoQuest questionnaire [22].

## 2. Materials and Methods

### 2.1. Design

We conducted a cross-sectional study to investigate the prevalence of misophonia.

### 2.2. Participants

This study enrolled students from technical and undergraduate programs at the Federal Institute of Education, Science, and Technology in Goias, Brazil. The technical courses at the Federal Institute are offered in two modalities: integrated with high school education and in the form of Adult and Youth Education (AYE). The first modality comprises young individuals aged between 13 and 18 years, while the second includes adults aged over 18 years. The research team visited students in their respective classes, inviting them to participate in the study. During these visits, students were informed about the research objectives and the procedures for accessing and completing the online questionnaire. Additionally, it was emphasized that minors were required to obtain parental consent through the signing of the Informed Assent Form before accessing the questionnaire. Students with physical, intellectual limitations, or known mental disorders were excluded from this study. Undergraduate students (over 18 years old) from various courses at the federal institute were contacted, invited, and instructed about the research through course and class groups. All participants completed a Free and Informed Consent Form signed by their parents/legal guardians and individually signed the Informed Consent Form.

In total, this study enrolled 549 students from both technical and undergraduate programs at the Federal Institute of Education, Science, and Technology in Goias, Brazil. Participants included a diverse group of students, with technical course participants aged between 13 and 18 years for those in the integrated high school modality, and over 18 years in the Adult and Youth Education (AYE) modality. Undergraduate participants were all over 18 years old. Exclusion criteria included individuals with physical, intellectual limitations, or known mental disorders. The gender distribution and additional demographic information, including names and email addresses, were collected as part of the online assessment process to facilitate follow-up and analysis. This approach ensured a representative and ethical inclusion of participants in line with this study’s objectives, as approved by the Research Ethics Committee (reference number: 5.631.371).

### 2.3. Online Assessments

The initial page of the form featured an exhaustive task description and researchers’ contact details. The data gathered encompassed the following information: name, email, gender, and age. The MisoQuest was made available to participants during the period between 15 September 2022 and 15 August 2023.

The participants completed the MisoQuest [22] questionnaire, a deliberate choice based on its distinction as a singular, fully validated misophonia questionnaire. The MisoQuest, a recently developed self-report questionnaire, has been demonstrated to possess robust psychometric properties, making it an ideal tool for assessing the prevalence of misophonia. The instrument exhibits high internal consistency, with Cronbach’s alpha values exceeding 0.90, indicating that its items are reliably measuring the construct of misophonia. Furthermore, the MisoQuest has shown strong test–retest reliability, ensuring stability and consistency of the results over time [22]. The MisoQuest is composed of 14 items, and each of the 14 items is rated on a scale of 1 to 5, representing varying degrees of agreement. The total score is derived by summing the scores for each item, yielding a range of 14–70. A total score equal to or exceeding 61 signifies a detection of misophonia [22]. The establishment of this cutoff by Siepsiak et al. [22] involved deducting one standard deviation (SD = 4.3) from the mean total score of individuals with misophonia (mean = 65.72). The Brazilian Portuguese version was derived from the original English version provided by the creators of the MisoQuest [22]. Two translations were conducted: one by the authors of the present study and another by a bilingual professional. Both translations were back-translated and compared with the original version. The MisoQuest was administered through online forms.

### 2.4. Statistical Analysis

Statistical analysis was conducted using the Python programming language [23] in conjunction with Jupyter Notebooks [24] for the development and execution of interactive scripts. Google Colab was utilized for conducting CFA using the lavaan and pwr packages from R. The primary statistical libraries employed included NumPy [25], Pandas [26], and SciPy [27]. These tools were integral for data manipulation, analysis, and visualization, enabling a comprehensive assessment of misophonia occurrence among students via the MisoQuest. A flowchart of the statistical analysis process is shown in Figure 1.

#### 2.4.1. Descriptive Statistics

Descriptive statistics, including means, standard deviations, frequencies, and percentages, were calculated to summarize the responses to the 14 items on the MisoQuest. These statistics provided a foundational understanding of the general trends and patterns within the dataset, offering insights into the distribution and central tendency of responses.

#### 2.4.2. Internal Consistency Reliability

The internal consistency reliability of the MisoQuest was assessed using Cronbach’s alpha. This test was performed to ensure that the items on the MisoQuest reliably measured the construct of misophonia. The Cronbach’s alpha values were found to exceed the acceptable threshold of 0.70, indicating a high level of internal consistency and reliability of the instrument.

#### 2.4.3. Factor Analysis

To validate the construct validity of the MisoQuest, confirmatory factor analysis (CFA) was conducted [22,28]. CFA was performed using the WLSMV (weighted least squares mean and variance) adjusted estimator, which is particularly suitable for categorical or ordinal data, such as Likert scales, and is robust against violations of normality. The model fit was evaluated using the ratio of chi-square to degrees of freedom $\chi^{2}/\mathrm{df}$), root mean square error of approximation (RMSEA), comparative fit index (CFI), and Tucker–Lewis index (TLI). The cutoff values used for assessing model fit were as follows: $\chi^{2}/\mathrm{df}\leq 3$, RMSEA ≤0.06, CFI ≥0.95, and TLI ≥0.95. These indices confirmed that the MisoQuest accurately measures the constructs related to misophonia.

### 2.5. Power Analysis

To ensure that the sample size used in CFA was adequate, a power analysis was conducted. Given 13 predictors (the items from the MisoQuest) and an effect size of $f^{2}=0.09$, the analysis indicated that, with a significance level of 0.05 and a desired power of 0.8, the residual degrees of freedom (*v*), which is determined by the difference between the number of observed variables (variances and covariances) and the number of estimated parameters, needed to be approximately 195.7. This corresponds to a required sample size of around 210 participants to reliably detect effects of small to moderate magnitudes. Considering that our actual sample size was 549, the analysis confirms that this study had sufficient power to detect hypothesized effects. This underscores the robustness and validity of the CFA results for both the one-factor model, which utilized a sample of 274 participants, and the three-factor model, which was based on a sample of 549 participants, as presented in this study [28,29].

## 3. Results

### 3.1. Misophonia Prevalence among Students via MisoQuest

A total of 549 students (mean = 23.2 years old, SD = 9.3) were enrolled in our study, comprising 285 females, 260 males, and 4 individuals who did not disclose their gender. Following the framework described by Siepziak et al. [22], we assessed misophonia using the MisoQuest tool, with scores ranging from 14 to 70 and a predefined cutoff value of 61 [14,22]. Considering this value, 4.5% of the participants were positive for misophonia. The MisoQuest score histogram is shown in Figure 2.

### 3.2. Adequacy of Data for Factor Analysis

To assess the suitability of the dataset for conducting exploratory factor analysis (EFA), Bartlett’s test of sphericity and the Kaiser–Meyer–Olkin (KMO) measure of sampling adequacy were performed.

Bartlett’s test of sphericity yielded a chi-square value of 1949.73 with a *p*-value of 0.001, indicating that the correlations between items were significantly different from zero and that the correlation matrix is not an identity matrix. This finding suggests that the data are suitable for factor analysis due to the presence of significant correlations among the items.

The Kaiser–Meyer–Olkin (KMO) measure of sampling adequacy for the overall dataset was found to be 0.94. According to Kaiser (1970) [30], a KMO value above 0.9 is classified as “excellent”, indicating that the sample is adequate for EFA. Furthermore, the KMO values for individual items ranged from 0.92 to 0.95, further confirming the sampling adequacy for each variable in the model. These results suggest that the data are highly suitable for factor analysis and provide a strong basis for subsequent statistical procedures in this study.

#### 3.2.1. Exploratory Factor Analysis (EFA)

We randomly divided our sample into two subsets and performed exploratory factor analysis (EFA) on the first subset (n = 275) and confirmatory factor analysis (CFA) on the second subset (n = 274). EFA was conducted to identify the underlying factor structure of the MisoQuest. According to Kaiser’s criterion, only factors with eigenvalues greater than 1 should be retained. As shown in the scree plot in Figure 3, there is a clear inflection point after the first factor, indicating that the first factor explains the majority of the variance in the data. Therefore, a single-factor model was initially deemed appropriate for the MisoQuest, as suggested by Siepziak et al. [22]. However, further analysis and recent studies have suggested that a multi-factor model may better capture the complexity of the construct measured by the MisoQuest, highlighting the need for both EFA and CFA to thoroughly assess the instrument’s dimensionality.

#### 3.2.2. Factor Loadings and Communalities for MisoQuest Items

The results of exploratory factor analysis (EFA) for the MisoQuest are presented in Table 1, showing the factor loadings and communalities for each item (Q1 to Q14).

The factor loadings range from 0.600 (Q2) to 0.792 (Q12). In general, factor loadings above 0.6 are considered moderate to high, indicating that the items have a substantial association with the underlying factor. In this case, all factor loadings are well above the commonly accepted minimum threshold of 0.3, suggesting that all items are good measures of the common latent factor.

The communalities range from 0.360 (Q2) to 0.627 (Q12). Communality represents the proportion of an item’s variance that can be explained by the extracted factors. Values above 0.5 are considered acceptable to indicate that the item is representative of the latent factor. Most items have communalities close to or above this value, suggesting that most items of the MisoQuest are well represented by the extracted factor.

These results indicate that the MisoQuest exhibits a well-defined latent factor, with all items contributing significantly to measuring this factor. This supports previous findings of high internal consistency and structural validity of the instrument in previous studies.

#### 3.2.3. Confirmatory Factor Analysis (CFA)

To confirm the single-factor structure identified in EFA, an initial round of confirmatory factor analysis (CFA) was conducted using the WLSMV estimator. This estimator is particularly suitable for categorical or ordinal data (such as Likert scales) and is robust against violations of normality. WLSMV adjusts the standard errors and chi-square statistics to account for non-normality and the categorization of data [31]. The factor loadings for all observed variables (Q1 to Q14) were significant, with *p*-values very close to zero, indicating a strong association between all variables and the latent factor F1. The standardized loadings ranged from 0.39 (Q10) to 0.78, suggesting that all variables contribute significantly to the latent factor.

The residual variances (e.g., Q1–Q1, Q2–Q2, etc.) were all positive and significant, which is expected, but this also indicates that a substantial amount of variation is not explained by the latent factor. This is common but may suggest the presence of additional factors or some heterogeneity in the responses. However, the fit indices, specifically the comparative fit index (CFI) and Tucker–Lewis index (TLI), were both above 0.95 (CFI = 1.00, TLI = 1.00), suggesting an excellent model fit. The RMSEA (root mean square error of approximation) value was less than 0.001, indicating a perfect fit, which is unusual and may indicate overfitting, particularly if the sample is small or the model is too simple. The SRMR (standardized root mean square residual) was 0.046, which is below the threshold of 0.08, indicating a good fit. The obtained value of $\frac{\chi^{2}}{df}<1.0$ is well within the acceptable range (≤3.0), indicating an excellent model fit to the observed data.

Despite the initial results, a second round of CFA was conducted to examine a three-factor structure for the MisoQuest, as proposed by Ay et al. (2024). The first factor (1 to 5 and 7 items) was named “Emotional Reactions”, the second factor (6, 8, 9, 12, 14 items) was named “Anger and Avoidance”, and the third factor (10, 11, 13 items) was named “Functionality” [28]. The corrected item–total correlation coefficients ranged from 0.55 to 0.74, as detailed in Table 2 for each individual item.

The goodness-of-fit indices for this model were highly favorable, with CFI = 1.00, TLI = 1.00, RMSEA = 0.001, and SRMR = 0.037. Additionally, the model fit was evaluated using the ratio of chi-square to degrees of freedom ($\chi^{2}/df$). The obtained value of $\frac{\chi^{2}}{df}=1.0$ is well within the acceptable range (≤3.0), indicating an excellent model fit to the observed data. These results indicate that there were no statistical differences between the single-factor and three-factor models, suggesting that either model could potentially fit the data well (Figure 4).

However, given the theoretical support for a multi-factor structure, further investigation may be warranted to determine the most appropriate model.

#### 3.2.4. Reliability Analysis

Cronbach’s alpha was calculated to assess the internal consistency of the MisoQuest. The resulting alpha coefficient was 0.93, indicating excellent reliability. This high value suggests that the items on the MisoQuest are highly inter-related and measure the same underlying construct.

## 4. Discussion

The present study investigated the prevalence and psychometric properties of the MisoQuest in a Brazilian sample of high school and undergraduate students, revealing insightful findings on the assessment of misophonia. To the best of our knowledge, this study is the first to evaluate the prevalence of misophonia among high school and undergraduate students in South America using a validated questionnaire. Among the 549 participants—comprising 285 females, 260 males, and 4 individuals who did not disclose their gender—4.5% were identified as positive for misophonia based on the MisoQuest score with a cutoff value of 61, as recommended by Siepsiak et al. (2020) [22].

To date, MisoQuest has been used in three prevalence studies, each differing in sample characteristics and results. Siepziak et al. (2020) investigated the prevalence of misophonia in 94 patients with depression using a face-to-face interview and the MisoQuest, finding a prevalence rate of 8.5% [14]. Enzler et al. (2021) explored misophonia prevalence in 253 individuals over 18 years old recruited from a list of emails and social network groups identified as misophonic or with other auditory sensitivities, reporting a prevalence of 45% [32]. Savard et al. (2022) studied a sample of 300 individuals from an online community, with no known history of misophonia or any other auditory sensitivity. They found that less than 2% of the subjects were positive for misophonia [33]. Our results align more closely with those of Savard et al. (2022). Our sample consisted of 549 students with no prior history of misophonia or any known or diagnosed hearing disorders.

Although the MisoQuest has demonstrated good psychometric properties [22,28,32], with a high specificity in correctly identifying those without misophonia, it exhibits low sensitivity in accurately identifying individuals with misophonia [14,32]. This suggests that the occurrence of false negatives may be high in samples similar to those of Savard et al. and the present study. The low sensitivity of the MisoQuest likely explains the findings in our study. Moreover, the authors of the MisoQuest have acknowledged the need for a validation of the psychometric properties of this instrument in populations akin to those in the present study [22].

Previous studies on the prevalence of misophonia in similar populations, but using different psychometric instruments, have reported varying results. In separate investigations, Wu et al. [34] and Zhou et al. [17] conducted surveys to determine the prevalence of misophonia among American and Chinese undergraduate students, respectively, both reporting identical prevalence rates of 20% within these populations. Similar results were found by Brennan et al. (2023) [21]. Another study conducted in the general population of Germany using MQ revealed a prevalence of 5% for misophonia symptoms among adults [15]. Unfortunately, these studies utilized the Misophonia Questionnaire (MQ), a tool distinct from that employed in our investigation, rendering direct comparisons between our results and theirs unfeasible.

Similarly, Naylor et al. [13] utilized the Amsterdam Misophonia Scale (A-Miso-S) to investigate the prevalence of misophonia among undergraduate medical students, reporting a prevalence rate of 49.1%. Other studies employing the A-Miso-S among young students, such as those conducted by Sarigedik and Gule (2021) [7], Aryal and Prabhu [35], and Sujeeth et al. (2024) [36], found prevalence rates of 13.8%, 48.3%, and 34.7%, respectively. Although these findings are similar to those reported by Enzler et al. [32], they are not directly comparable to the results of our study due to the use of different assessment instruments. In this direction, Kula et al. [37] conducted a comprehensive evaluation of the psychometric properties of current instruments for hyperacusis and misophonia using the COSMIN guidelines. Their study demonstrated that, according to psychometric criteria, the MisoQuest achieved sufficient structural validity and internal consistency. In contrast, the MQ was deemed indeterminate for both criteria, whereas A-Miso-S was found to have insufficient structural validity but sufficient internal consistency. Notably, Kula et al. [37] presented compelling evidence that the MisoQuest stood out as the sole measurement instrument yielding reliable results, as evidenced by the measured interclass correlation coefficient (ICC). These findings further support the assertion that results obtained using different tools are not consistently comparable. Additionally, the authors of the MisoQuest did not conduct a comparative analysis with the MQ and A-MISO-S due to methodological issues and diagnostic criteria that make these instruments unsuitable for testing the convergent validity of the MisoQuest [22].

Regarding the psychometric evaluation of MisoQuest’s Brazilian version, the results of Bartlett’s test of sphericity and the Kaiser–Meyer–Olkin (KMO) measure of sampling adequacy confirmed that the data are suitable for factor analysis. With a chi-square value of 1949.73 and a highly significant *p*-value (<0.001), the Bartlett test indicates substantial correlations between items, rejecting the hypothesis that the correlation matrix is an identity matrix. The KMO value of 0.94, classified as “excellent” according to Kaiser (1970), reinforces the adequacy of the data for factor analysis, which is essential for validating the dimensional structure of the Brazilian version of the MisoQuest. These results align with the literature, suggesting the use of preliminary tests to confirm the viability of EFA, ensuring that the observed correlations are not merely coincidental [22,28].

In exploratory factor analysis (EFA), the retention of a single factor, as indicated by Kaiser’s criterion and visualized in the scree plot, suggests that a one-factor model could initially explain a significant portion of the variance in the data. However, recent studies, including the study by Ay et al. (2024) [28], point to the possibility of a multifactorial structure, which motivated the subsequent confirmatory factor analysis (CFA) to be utilized to investigate the adequacy of alternative models. This methodological approach is crucial because it allows for the testing of theoretical and empirical hypotheses regarding the dimensionality of the instrument.

The factor loadings of items in EFA ranged from 0.600 to 0.792, with all being well above the minimum threshold of 0.3, indicating a substantial association of the items with the underlying latent factor. The communalities of items ranged from 0.360 to 0.627, with most items reaching values close to or above 0.5, indicating that a significant proportion of the variance of each item is explained by the extracted factor. These results confirm the robustness of the Brazilian version of the MisoQuest in consistently measuring a well-defined latent construct, consistent with the high internal consistency previously reported for the instrument. However, it is important to note that item Q2 presented the lowest communality (0.360), suggesting that although it contributes to the common factor, its representativeness may be limited compared to other items.

The initial CFA, using the WLSMV estimator for ordinal data, confirmed the unifactorial model identified in EFA, with all standardized loadings being significant and high fit indices (CFI and TLI both equal to 1.00, RMSEA close to zero). However, these results indicate an unusually perfect fit, which may suggest overfitting, especially if the sample size is small or the model is overly simplified. To address this limitation, a second round of CFA examined a trifactorial model proposed by Ay et al., which reflected a more complex structure of the MisoQuest. The fit indices for this model were also excellent (CFI and TLI = 1.00, RMSEA = 0.037), indicating that both the unifactorial and trifactorial models fit well to the data.

Despite the lack of a statistically significant difference between the unifactorial and trifactorial models, the choice of the most suitable model should consider the balance between the simplicity of the model and its ability to capture the theoretical complexity of the construct. The high corrected item–total correlations (ranging from 0.55 to 0.74) suggest that all items are valuable contributors to the identified factors. The decision to adopt a multifactorial model could be supported by future analyses, which could explore the stability and predictive validity of these factors in larger and more diverse samples.

Finally, reliability analysis revealed a Cronbach’s alpha value of 0.93, indicating excellent internal consistency for MisoQuest items. This result reinforces that the MisoQuest questionnaire is a reliable instrument for measuring misophonia, with highly correlated items capturing a cohesive construct [22,28].

Additionally, Siepsiak et al. (2020) [22] demonstrated that early identification and intervention are crucial in managing misophonia symptoms. Thus, providing appropriate support and accommodations in educational settings can significantly enhance the quality of life for affected individuals, including students.

## 5. Conclusions

In conclusion, our study contributes to the growing body of literature on misophonia by validating the psychometric properties of the Brazilian version of the MisoQuest in a sample of high school and undergraduate students. The findings confirm that the MisoQuest is a reliable tool for identifying misophonia, with robust factor loadings and high internal consistency, supporting both unifactorial and multifactorial models. However, the exploration of a trifactorial model suggests that the construct of misophonia may be more complex than previously thought, indicating potential areas for refinement in its measurement model.

Given the prevalence and impact of misophonia among young adults, particularly in academic settings, it is essential to continue research to develop effective assessment and intervention strategies. Future research should further investigate the underlying dimensions of misophonia to enhance our understanding and assessment of this condition, which may include testing the stability and predictive validity of the identified factors in larger and more diverse samples. Such efforts could ultimately improve support and accommodations for those affected by misophonia, enhancing their quality of life and educational experiences.

## 6. Limitations

This study provides valuable insights into the prevalence of misophonia among high school and undergraduate students in Brazil and evaluates the psychometric properties of the MisoQuest in this population. However, several limitations must be acknowledged.

Firstly, the reliance on self-reported data, particularly through the use of the MisoQuest, introduces potential biases, such as social desirability and recall bias. While the MisoQuest has been validated in prior studies and demonstrates high specificity, its low sensitivity may lead to an underestimation of misophonia prevalence. The potential for false negatives, particularly in populations without a known history of misophonia or related auditory sensitivities, suggests that the true prevalence may be higher than reported.

Secondly, this study’s cross-sectional design limits the ability to infer causality or track changes in misophonia prevalence over time. Longitudinal studies would be beneficial in understanding the progression of misophonia symptoms and their potential impact on individuals’ academic and social functioning.

Another limitation is the lack of direct comparability with other studies that employed different assessment tools, such as the Misophonia Questionnaire (MQ) or the Amsterdam Misophonia Scale (A-Miso-S). Although the MisoQuest was selected for its favorable psychometric properties, including sufficient structural validity and internal consistency—as demonstrated by Kula et al. (2023) [37], differences in diagnostic criteria and assessment methods across studies hinder a consistent comparison of prevalence rates.

Additionally, the sample composition, consisting solely of high school and undergraduate students, may limit the generalizability of the findings to other age groups or populations with different educational backgrounds. Further research should aim to include more diverse samples to better understand the prevalence and impact of misophonia across various demographic and cultural contexts.

Finally, the statistical validation of the MisoQuest in this study was robust, with high reliability (Cronbach’s alpha = 0.93) and favorable fit indices in both the single-factor and three-factor confirmatory factor analyses (CFAs). However, the perfect fit indices observed (e.g., CFI = 1.00, TLI = 1.00, RMSEA = 0.001) raise concerns about potential overfitting, which may be a consequence of the model’s simplicity or the sample size. While the model fit was excellent, further validation in larger and more heterogeneous samples is necessary to confirm these findings and ensure the tool’s broader applicability.

In summary, while this study advances our understanding of misophonia prevalence and the psychometric validity of the MisoQuest, the limitations highlighted here underscore the need for a cautious interpretation of the results and suggest directions for future research.

## Figures and Tables

**Figure 1 healthcare-12-01888-f001:**
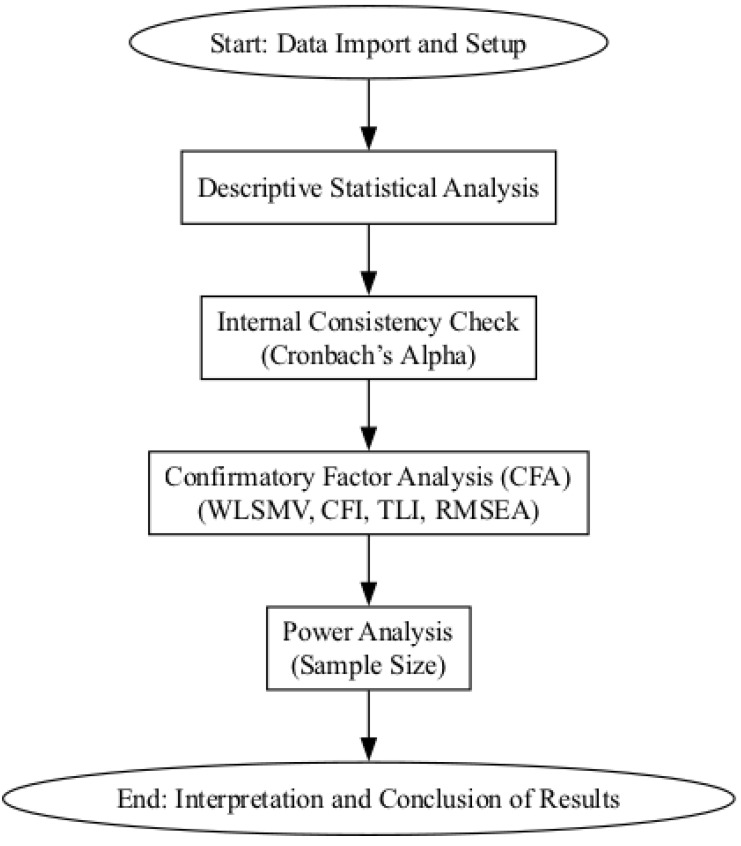
Flowchart of the statistical analysis process.

**Figure 2 healthcare-12-01888-f002:**
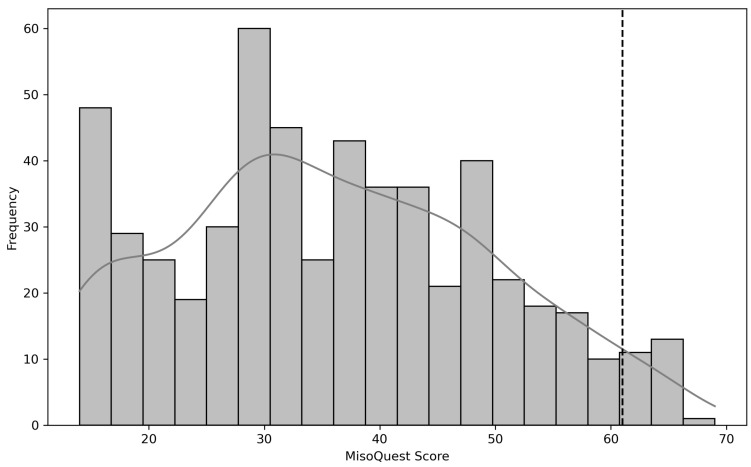
The MisoQuest score histogram. The MisoQuest cutoff (=61) for misophonia is shown with a black dotted line.

**Figure 3 healthcare-12-01888-f003:**
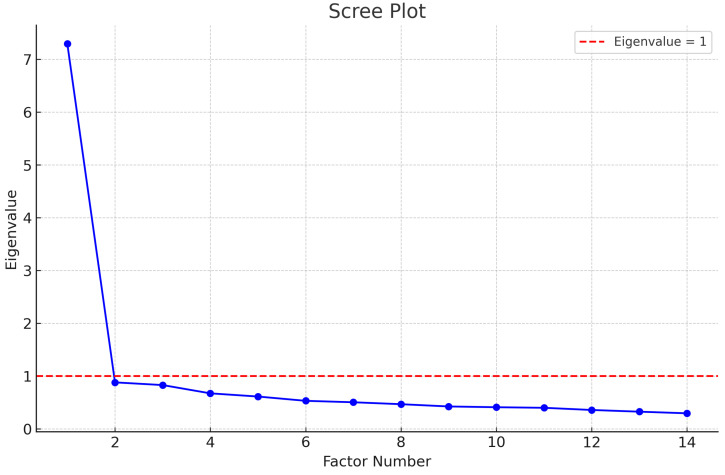
Scree plot of eigenvalues for the MisoQuest.

**Figure 4 healthcare-12-01888-f004:**
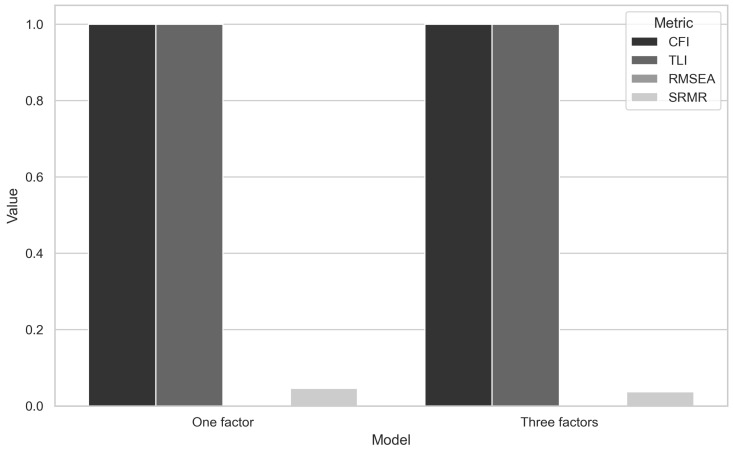
Comparative fit index across models.

**Table 1 healthcare-12-01888-t001:** Factor loadings and communalities for MisoQuest items.

Items	Factor Loadings	Communalities
Q1	0.717	0.514
Q2	0.600	0.360
Q3	0.730	0.533
Q4	0.772	0.597
Q5	0.717	0.515
Q6	0.711	0.505
Q7	0.757	0.573
Q8	0.708	0.502
Q9	0.696	0.484
Q10	0.742	0.551
Q11	0.779	0.607
Q12	0.792	0.627
Q13	0.699	0.489
Q14	0.769	0.591

**Table 2 healthcare-12-01888-t002:** Corrected item–total correlations and factor loadings for MisoQuest items.

MisoQuest Items (n = 549)	Corrected Item–Total Correlation	F1	F2	F3
Q1	0.644	0.691		
Q2	0.555	0.594		
Q3	0.677	0.726		
Q4	0.677	0.727		
Q5	0.643	0.686		
Q7	0.699	0.749		
Q6	0.664		0.712	
Q8	0.630		0.671	
Q9	0.658		0.700	
Q12	0.719		0.794	
Q14	0.694		0.762	
Q10	0.741			0.764
Q11	0.650			0.740
Q13	0.715			0.687

## Data Availability

All research data have been made available in the Appendix A.

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
