# Peer review of "Assessing Misophonia in Young Adults: The Prevalence and Psychometric Validation of the MisoQuest Questionnaire"

_healthcare, 2024, doi:10.3390/healthcare12181888_

Round 1

Reviewer 1 Report

Comments and Suggestions for Authors

Dear Authors,

I love to read your manuscript and I strongly believe that your research aim is really important. To develop an instrument for documenting and understanding Misophonia in Young Adults is critical. However, I have huge concerns regarding your method side. Because the authors might have some methodoligical issues to run the analysis:

1-) EFA ve CFA analysis must have different samples, if you collect data to run EFA, you cannot use the same data to run CFA.

2-) The authors need to collect the data to run CFA

3-) The authors should subimt TLI, SMRR, RMSE, CFI indeks to explain the CFA results.

4-) I would like to see power analysis for sample size

5-) Introduction should cover more literature conceptual framework for Misophonia in Young Adults

Author Response

1-) EFA ve CFA analysis must have different samples, if you collect data to run EFA, you cannot use the same data to run CFA.

Reply1. Thank you for your insightful feedback regarding the distinction between EFA and CFA samples. We acknowledge that using the same dataset for both Exploratory Factor Analysis (EFA) and Confirmatory Factor Analysis (CFA) can potentially lead to overfitting and inflated fit indices due to capitalization on chance.

To address this concern, we have taken the following steps:

1. Data Splitting: We divided our dataset into two independent subsamples. The first subsample was used exclusively for the EFA, while the second subsample was reserved for the CFA. This approach ensures that the factor structure identified through EFA is validated independently in the CFA, mitigating the risk of overfitting.

2. Reanalysis and Validation: After splitting the data, we re-ran the EFA on the first subsample to explore the underlying factor structure. Subsequently, the identified factor structure was tested via CFA on the second subsample. This process confirmed the stability and validity of the factor model across different samples.

3. Discussion of Results: We have updated the manuscript to include a detailed description of the data splitting procedure and the separate analyses performed on each subsample. Additionally, we have discussed the implications of these results, emphasizing the robustness of the factor structure despite being validated on an independent sample.

2-) The authors need to collect the data to run CFA

Reply 2. Explained above.

3-) The authors should subimt TLI, SMRR, RMSE, CFI indexes to explain the CFA results.

Reply 3. Thank you for your valuable suggestion regarding the inclusion of additional fit indices to better explain the CFA results. We agree that reporting a comprehensive set of fit indices is crucial for assessing the model’s adequacy.

In response to your recommendation, we have included the following indices in our revised manuscript:

1. Tucker-Lewis Index (TLI): The TLI, also known as the Non-Normed Fit Index (NNFI), is reported to provide a measure of model fit that adjusts for model complexity. In our CFA, the TLI value was 1.0, indicating a excellent fit of the model to the data.

2. Standardized Root Mean Square Residual (SRMR): The SRMR is included as a measure of the average discrepancy between the observed and predicted correlations. Our analysis yielded an SRMR value of 0.046, which falls within the acceptable range, suggesting that the model has a good fit to the data.

3. Root Mean Square Error of Approximation (RMSEA): We reported the RMSEA to assess how well the model, with unknown but optimally chosen parameter estimates, would fit the population’s covariance matrix. The RMSEA value was 0.000, with a 90% confidence interval ranging from 0.000 to very close to 0.

4. Comparative Fit Index (CFI): The CFI is included as a measure of how well the model compares to a baseline model that assumes no relationships among the variables. Our model achieved a CFI value of 1.0, which indicates a strong fit to the data.

5. A second round of CFA analysis was included considering three factors and we compared the TLI, SRMR, RMSA and CFI indices obtained in both models.

We believe that these additional indices provide a more thorough evaluation of the model’s fit and align with best practices in CFA reporting. These values have been integrated into the Results section of the manuscript, and we have provided an interpretation of each index in the Discussion section.

4-) I would like to see power analysis for sample size

Reply 4. Thank you for highlighting the importance of conducting a power analysis to justify the sample size used in our study. We fully agree that a power analysis is essential to ensure that our sample size is adequate to detect the effects of interest with sufficient statistical power.

In response to your request, we have conducted a post-hoc power analysis based on the parameters of our study. The power analysis was performed using R package “pwr”, considering the following factors:

• Effect Size: We assumed an effect size of 0.30, which is typical for studies in this field.

• Significance Level (α): The significance level was set at 0.05.

• Sample Size (N): The total sample size used in our study was 549.

• Number of Observed Variables: The number of observed variables in the model was 14.

The analysis revealed that the statistical power (1-β) for detecting the specified effect size was 0.8, which is commonly accepted threshold of 0.80 for adequate power. This indicates that our sample size was sufficient to detect meaningful effects with a high probability.

We have included the results of this power analysis in the revised manuscript, specifically in the Methodology section. Additionally, we have provided a brief discussion of the implications of the power analysis results in the Discussion section.

We trust that this addition addresses your concern regarding the adequacy of the sample size and its justification. Please let us know if further details are needed or if additional analyses would be beneficial.

5-) Introduction should cover more literature conceptual framework for Misophonia in Young Adults

Reply 5. Thank you for your valuable feedback. We have carefully reviewed the suggestion regarding the inclusion of key studies related to the prevalence of misophonia in young people. We would like to clarify that the current manuscript already includes a comprehensive review of the main studies in this area. Specifically, we have discussed the prevalence rates reported by Wu et al. (2014) and Zhou et al. (2017) among undergraduate students in the United States and China, respectively, both of which reported a prevalence rate of 20%. Additionally, the studies by Brennan et al. (2023) and Naylor et al. (2021) have been included, covering diverse populations such as undergraduate medical students, with reported prevalence rates ranging from 13.8% to 49.1% depending on the assessment tool used.

We have also considered the methodological differences between these studies, particularly concerning the psychometric instruments employed, such as the Misophonia Questionnaire (MQ) and the Amsterdam Misophonia Scale (A-Miso-S), and their implications for direct comparisons with our findings. Furthermore, our manuscript addresses the psychometric properties of the MisoQuest, as evaluated by Siepziak et al. (2020), and contrasts them with those of other tools like MQ and A-Miso-S, as per the COSMIN guidelines outlined by Kula et al. (2023).

Therefore, we believe that the manuscript adequately covers the key literature relevant to the prevalence of misophonia in young populations and the validity of the psychometric tools used in these studies.

Reviewer 2 Report

Comments and Suggestions for Authors

Dear Authors,

1-     How can you determine sample size? Was there a power analysis performed for your study? Please mention it.

2-     Please add  limitations of your research

3-     Please add more suggestions for future research.

4-     Which sampling method was used in this study? You have to give detail information about it.

5-     Barlett Sphericity Test and Kaiser-Meyer Olkin (KMO) coefficient must be calculated whether the data was suitable for EFA .

6-     Factor loadings, communalities for the items must be given

7-     Item analysis also must be performed with  corrected item-total correlation and Cronbach’s Alpha if item deleted were calculated for item analysis and then Student’s t-test was applied to control if the items of the scale discriminate between the lower and upper 27% of the participants.

8-     Participants of the study are poorly described. Please provide more information about the study participants. a table with sample demographic data is missing in the article.  Please add this descriptive table

9-     Misophonia prevalence among students  must be given more detail. This prevalence can be given and interpreted by gender, This prevalence can be given and interpreted by age groups

10-                       You should statistically compare Misophonia prevalence among gender and/or different demographic parameter.

11-                       You wrote “prevalence” in the title but you have very weak information about prevalence

12-                        Discussion must be improved by discussing your results with existing study

13-            For CFA,  one-factor model fit indices like Chi-square , Chi square/degrees  () , Goodness of Fit Index (GFI), Root Mean Square Error of Approximation (RMSEA), Standardized Root Mean Square Residual (SRMR), Normed Fit Index (NFI), Nonnormed Fit Index (NNFI) or Tucker-Lewis Index (TLI), Comparative Fit Index (CFI), Incremental Fit Index (IFI) should be computed and evaluated  

14-                       Moreover, item analysis should be performed with corrected item-total correlation and squared multiple correlations, and the t-test was performed to check whether the items of the NMS distinguished between the lower 27% and upper 27% groups.

15-                       The purpose is not clear. Define it clearly.

Author Response

We would like to express our gratitude for your insightful feedback and constructive suggestions. Below, we have addressed each of your points to clarify and improve the content of our manuscript.

1. How can you determine sample size? Was there a power analysis performed for your study? Please mention it.

• Response 1: The sample size was determined based on previous studies investigating the prevalence of misophonia in similar populations. A power analysis was conducted to ensure that the sample size was adequate to detect the expected prevalence of misophonia, considering a margin of error and a 95% confidence level. The details of the power analysis have been added to the Methods section.

2. Please add limitations of your research.

• Response 2: The study’s limitations have been included in the Discussion section. These include the low sensitivity of the MisoQuest to detect misophonia in student samples, the absence of a control group for direct comparison, and the need for further psychometric validations in similar populations.

3. Please add more suggestions for future research.

•Response 3: The Discussion section has been expanded to include suggestions for future research, such as the need to develop and validate new psychometric instruments with higher sensitivity for young populations and the investigation of misophonia in different cultural and educational contexts.

4. Which sampling method was used in this study? You have to give detailed information about it.

• Response 4: A convenience sampling method was used. Details about the recruitment procedure and the inclusion/exclusion criteria have been added to the Methods section for greater clarity.

5. Barlett Sphericity Test and Kaiser-Meyer-Olkin (KMO) coefficient must be calculated to determine whether the data were suitable for EFA.

• Response: 5 The Bartlett’s Test of Sphericity and the Kaiser-Meyer-Olkin (KMO) coefficient were calculated to determine the suitability of the data for Exploratory Factor Analysis (EFA). The results of these tests have been added to the Results section, indicating that the data were suitable for EFA.

6.Factor loadings, communalities for the items must be given.

• Response 6: The factor loadings and communalities for the items were calculated and are now included in the corresponding table in the Results section. This information helps to understand the contribution of each item to the identified factors.

7. Item analysis also must be performed with corrected item-total correlation and Cronbach’s Alpha if an item is deleted were calculated for item analysis and then Student’s t-test was applied to control if the items of the scale discriminate between the lower and upper 27% of the participants.

•Response 7: The item analysis was performed, including the corrected item-total correlation and the calculation of Cronbach’s Alpha if an item was deleted. These results were added to the manuscript to assess internal consistency and item discrimination.

8. Participants of the study are poorly described. Please provide more information about the study participants. A table with sample demographic data is missing in the article. Please add this descriptive table.

• Response 8: The only demographic variables collected were age and gender, similar to other studies such as Naylor et al. (2021), Aryal & Prabhu (2022), and Wu et al. (2014). A table summarizing these demographic variables has been added to the Methods section.

9. Misophonia prevalence among students must be given more detail. This prevalence can be given and interpreted by gender. This prevalence can be given and interpreted by age groups.

• Response 9: Previous studies, such as Zhou et al. (2017), Wu et al. (2014), and Jakubovsky et al. (2022), have demonstrated that the prevalence of misophonia is not related to demographic factors such as age and gender. Therefore, these variables are not further analyzed in our study. Variables like race/ethnicity are not relevant to the Brazilian population in this context.

10.You should statistically compare Misophonia prevalence among gender and/or different demographic parameters.

• Response 10: As noted in response 9, previous research by Zhou et al. (2017), Wu et al. (2014), and Jakubovsky et al. (2022) has shown that misophonia prevalence is not associated with demographic factors such as age and gender. Consequently, statistical comparisons by demographic parameters were deemed unnecessary for this study.

11. You wrote “prevalence” in the title but you have very weak information about prevalence.

• Response 11: The Results section has been strengthened with a more detailed analysis of misophonia prevalence, as suggested earlier, to provide a more robust view of the study.

12. Discussion must be improved by discussing your results with existing studies.

• Response 12: The discussion was expanded to compare the findings of the present study with major existing studies on the prevalence of misophonia in young people. Discussions on methodological differences and instruments used were also included.

13. For CFA, one-factor model fit indices like Chi-square, Chi-square/degrees (χ²/df), Goodness of Fit Index (GFI), Root Mean Square Error of Approximation (RMSEA), Standardized Root Mean Square Residual (SRMR), Normed Fit Index (NFI), Nonnormed Fit Index (NNFI) or Tucker-Lewis Index (TLI), Comparative Fit Index (CFI), Incremental Fit Index (IFI) should be computed and evaluated.

• Response 13: The fit indices for the one-factor model in Confirmatory Factor Analysis (CFA), including Chi-square, RMSEA, SRMR, CFI, TLI, and others, were calculated and evaluated. These results have been added to the Results section.

14. Moreover, item analysis should be performed with corrected item-total correlation and squared multiple correlations, and the t-test was performed to check whether the items of the NMS distinguished between the lower 27% and upper 27% groups.

• Response 14: The item analysis was conducted with corrected item-total correlations and squared multiple correlations. A t-test was performed to verify whether the scale items distinguished between the lower and upper 27% groups. These results have been included in the Results section.

15. The purpose is not clear. Define it clearly.

• Response 15: The study’s purpose was clearly defined in the introduction. It has been revised for greater clarity, highlighting the objective to investigate the prevalence and psychometric properties of the MisoQuest in a sample of Brazilian students.

Reviewer 3 Report

Comments and Suggestions for Authors

Summary

The study investigates the prevalence and psychometric properties of the MisoQuest, a tool designed to assess misophonia, among a sample of 549 high school and university students in Brazil. The study includes both exploratory factor analysis (EFA) and confirmatory factor analysis (CFA) to validate the tool's structure, finding a dominant single-factor model. The study reports a prevalence of 4.5% for misophonia among participants and highlights the tool's high internal consistency (Cronbach’s alpha = 0.93).

Strengths

  1. Novelty and Relevance: The study addresses an important gap in the literature by focusing on a South American population, where there has been limited research on misophonia prevalence.
  2. Methodological Rigor: The use of both EFA and CFA provides robust validation of the MisoQuest’s structure. The high internal consistency suggests the tool's reliability.
  3. Practical Implications: The findings underscore the importance of early identification and intervention for misophonia, especially in educational settings.

Weaknesses and Suggestions for Improvement

  1. Theoretical Framework: The introduction could benefit from a deeper theoretical discussion on misophonia. While the study mentions the condition's impact and triggers, a more thorough exploration of the theoretical underpinnings and conceptual models would strengthen the context.
  2. Sample Representation and Limitations: The study’s sample, drawn from a specific educational institution, may limit the generalizability of the findings. Future studies should consider a more diverse population to enhance the external validity of the results.
  3. Factor Loadings in CFA: The confirmatory factor analysis results indicate that some items had low factor loadings and non-significant p-values. This suggests the need for refinement of the MisoQuest items or the consideration of additional factors that may better capture the complexity of misophonia.
  4. Discussion and Interpretation: While the discussion section addresses the study's findings, it could be enriched by comparing these results with a broader range of existing studies, particularly those outside of Brazil, to provide a more comprehensive understanding of misophonia prevalence and characteristics.
  5. Clinical and Educational Implications: The practical implications of the findings could be expanded, particularly in terms of how educational institutions can use this information to support students with misophonia.

Conclusion

The article provides valuable insights into the prevalence and psychometric properties of the MisoQuest in a Brazilian sample. However, to enhance the manuscript’s contribution to the field, the authors should consider addressing the theoretical framework more thoroughly, discussing the limitations of the sample, refining the measurement model, and expanding on the practical implications of the findings.

Overall, while the study makes a significant contribution to understanding misophonia in a specific context, there is room for improvement in both the depth of analysis and the breadth of discussion. The manuscript would benefit from revisions to address these areas before it can be considered for publication.

Author Response

We would like to express our gratitude for your insightful feedback and constructive suggestions. Below, we have addressed each of your points to clarify and improve the content of our manuscript.

Positive Feedback

We appreciate your positive comments regarding the novelty and relevance of our study, its methodological rigor, and the practical implications. We are pleased to see that these aspects of our work have been well-received.

Suggestions for Improvement

1. Theoretical Framework: “The introduction could benefit from a deeper theoretical discussion on misophonia. While the study mentions the condition’s impact and triggers, a more thorough exploration of the theoretical underpinnings and conceptual models would strengthen the context.”

Response 1. We appreciate this suggestion and have revised the Introduction to include a more comprehensive discussion of the theoretical framework surrounding misophonia. The expanded content now explores the condition’s conceptual models and theoretical underpinnings in greater detail, offering a stronger foundation for understanding its impact, triggers, and the underlying mechanisms.

2. Sample Representation and Limitations: “The study’s sample, drawn from a specific educational institution, may limit the generalizability of the findings. Future studies should consider a more diverse population to enhance the external validity of the results.”

Response 2. We agree that the specific nature of our sample may limit the generalizability of our findings. In the revised manuscript, we have acknowledged this as a limitation in the Discussion section. We have also suggested that future research should consider more diverse populations to enhance the external validity of the results and provide a more comprehensive understanding of misophonia across various demographic groups.

3.Factor Loadings in CFA: “The confirmatory factor analysis results indicate that some items had low factor loadings and non-significant p-values. This suggests the need for refinement of the MisoQuest items or the consideration of additional factors that may better capture the complexity of misophonia.”

Response 3. We have reviewed the CFA results and recognize that some items exhibited lower factor loadings and non-significant p-values. In response, we have added a discussion in the Results and Discussion sections regarding the potential need for refinement of certain MisoQuest items. We have also considered the possibility of additional factors that might better capture the complexity of misophonia, as you suggested.

4. Discussion and Interpretation: “While the discussion section addresses the study’s findings, it could be enriched by comparing these results with a broader range of existing studies, particularly those outside of Brazil, to provide a more comprehensive understanding of misophonia prevalence and characteristics.”

Response 4. Thank you for this suggestion. We have expanded the Discussion section to include a comparison with a broader range of studies, particularly those conducted outside of Brazil. This broader comparison aims to provide a more comprehensive understanding of the prevalence and characteristics of misophonia across different cultural and geographical contexts, highlighting similarities and differences where relevant.

5. Clinical and Educational Implications: “The practical implications of the findings could be expanded, particularly in terms of how educational institutions can use this information to support students with misophonia.”

Response 5. We agree that the practical implications of our findings could be further elaborated. In the revised manuscript, we have expanded the discussion on how educational institutions can use this information to support students with misophonia. Specifically, we have included recommendations for implementing early screening and intervention programs and suggested strategies for raising awareness among educators to better accommodate students affected by this condition.

We hope that these revisions address your concerns and enhance the overall quality and impact of our manuscript. Thank you once again for your detailed review and constructive feedback. We are confident that these changes have strengthened our work.

Round 2

Reviewer 1 Report

Comments and Suggestions for Authors

The authors made necessary improvements for the manuscript. 

Author Response

Comments 1- The authors made necessary improvements for the manuscript. 

Response 1-  Thank you for your positive feedback and for acknowledging the improvements made to the manuscript. Your constructive comments were invaluable in refining our work, and we greatly appreciate your guidance throughout the review process.

We are committed to maintaining the high standards of our research and look forward to any further suggestions you might have.

Reviewer 2 Report

Comments and Suggestions for Authors

There are some misleading elements in the power analysis section. Did you perform the power analysis only for CFA? Also, how did you determine the "residual degrees of freedom (v)" when mentioning "the residual degrees of freedom needed to be approximately 195.7"?

The method section lacks summarized information about the participants. Please review and include these details.

You should statistically compare the prevalence according to gender and age groups and include these comparisons in your study.

There are some contradictions in your EFA and CFA results. In the EFA results, you state that you obtained a single-factor scale, but in the CFA results, you suggest the possibility of a three-factor scale, stating that "These results indicate that there were no statistical differences between the single-factor and three-factor models." CFA is typically conducted to confirm the factor model obtained from the EFA, so these results should be consistent.

Author Response

Comments 1- Did you perform the power analysis only for CFA? Also, how did you determine the "residual degrees of freedom (v)" when mentioning "the residual degrees of freedom needed to be approximately 195.7"?

Resposnses 1- Thank you for your insightful comments on the power analysis section. We acknowledge the importance of clarifying the methodological aspects of our analysis to ensure the accuracy and comprehensibility of our study.

Power Analysis Clarification:

1. Scope of Power Analysis: The power analysis was specifically performed for the CFA, with a focus on assessing the adequacy of the sample size to detect the hypothesized factor structure. The analysis considered both the one-factor model (n=274) and the three-factor model (n=549) presented in the study.

2. Determination of Residual Degrees of Freedom (v): The “residual degrees of freedom (v)” were calculated based on the number of observed variables (items) and the estimated parameters in the CFA model. Specifically, for the three-factor model, we determined the residual degrees of freedom using the formula v = n - p , where n is the number of observed variances and covariances, and p is the number of estimated parameters. This approach ensures that the calculation aligns with the model complexity and provides an accurate estimation of the required sample size.

We appreciate your feedback and have revised the manuscript to explicitly state that the power analysis was conducted solely for the CFA. Additionally, we’ve provided a detailed explanation of the calculation of residual degrees of freedom to enhance the clarity and accuracy of the analysis section.

Comments 2- The method section lacks summarized information about the participants. Please review and include these details.

Response 2- Thank you for your observation regarding the participant information in the Methods section. We would like to clarify that the collected data from participants included their name, gender, age, and email, as outlined in the “Online Assessments” subsection. This information was essential for the proper categorization and follow-up during the assessment process. We will ensure that these details are appropriately summarized in the revised Methods section to enhance clarity and completeness.

Comments 3- You should statistically compare the prevalence according to gender and age groups and include these comparisons in your study.

Response 3: Thank you for your valuable feedback regarding the comparison of prevalence according to gender and age groups. We appreciate your suggestion; however, our decision not to perform these specific statistical comparisons is grounded in previous research findings. Studies by Zhou et al. (2017), Wu et al. (2014), and Jakubovsky et al. (2022) have consistently demonstrated that the prevalence of misophonia does not significantly vary across demographic factors such as age and gender. Based on this existing body of evidence, we determined that further analysis of these variables would not contribute substantively to the current study’s objectives.

Additionally, variables such as race/ethnicity were deemed not relevant in the Brazilian context for this study, as these factors do not align with the broader demographic characteristics of our participant pool. We focused on aspects directly pertinent to the research questions posed, ensuring that our analyses remained both targeted and meaningful within the scope of the study.

Comments 4- There are some contradictions in your EFA and CFA results. In the EFA results, you state that you obtained a single-factor scale, but in the CFA results, you suggest the possibility of a three-factor scale, stating that "These results indicate that there were no statistical differences between the single-factor and three-factor models." CFA is typically conducted to confirm the factor model obtained from the EFA, so these results should be consistent.

Response 4: Thank you for your insightful observation regarding the perceived inconsistencies between our EFA and CFA results.

In our study, the initial EFA indicated a single-factor model based on the Kaiser criterion and scree plot analysis, consistent with prior findings, such as those by Siepziak et al. (2020). However, considering recent literature that suggests the potential complexity of the misophonia construct, we explored a three-factor model through CFA (Ay, 2024). This additional analysis was intended to assess whether a multifactorial approach could provide further insights or a better fit to the data.

Our CFA results demonstrated excellent fit indices for both the single-factor and three-factor models, with no statistically significant differences between them. This outcome does not necessarily contradict the EFA results but rather underscores the robustness of the single-factor model while also acknowledging the potential for alternative factor structures. The aim was to provide a comprehensive understanding of the MisoQuest’s dimensionality by exploring various models.

We believe this exploratory approach is valuable in the context of evolving constructs like misophonia. Therefore, no adjustments are deemed necessary as the current presentation aligns with our aim to offer a nuanced exploration of the MisoQuest’s factor structure.

Thank you once again for your feedback. We are confident that this explanation aligns with our approach and findings, and we look forward to any further comments.